# Peer review of "Real-Time Automatic Investigation of Indian Roadway Animals by 3D Reconstruction Detection Using Deep Learning for R-3D-YOLOv3 Image Classification and Filtering"

_electronics, doi:10.3390/electronics10243079_

Round 1

Reviewer 1 Report

  1. Results: Recommend to be Major revisions

This paper briefs a detailed forum on GPU-based embedded systems and object detection applications. It provides a unique and real-time solution using DL-based Real- 3D Motion-based YOLOv3 (R-3D-YOLOv3) object detection and tracking of images on mobility. Besides, it discovers methods for multiple views of flexible objects using 3D reconstruction, especially for stray animals. It seeks solutions by forecasting Image Filters to find object properties and semantics for object recognition algorithms leading to closed-loop object detection and tracking.

This paper is with some merits for Electronics, however, it requires some major revisions.

Firstly, the abstract should be refined to clearly indicate what authors had done within 150 words. The abstract is saying nothing (lots of garbage wordings) to keep what authors had really done.

Secondly, for Sections 1 and 2, authors should provide the comments of the cited papers after introducing each relevant work. What readers require is, by convinced literature review, to understand the clear thinking/consideration why the proposed approach can reach more convinced results. This is the very contribution from authors. In addition, authors also should provide more sufficient critical literature review to indicate the drawbacks of existed approaches, then, well define the main stream of research direction, how did those previous studies perform? Employ which methodologies? Which problem still requires to be solved? Why is the proposed approach suitable to be used to solve the critical problem? We need more convinced literature reviews to indicate clearly the state-of-the-art development. And very importantly, authors always have to write a paragraph saying: “The rest of the paper is organized as follows. Section 2 contains the literature review. Section 3 contains the methodology (method). Section 4 contains the results. Section 5 contains the conclusions and policy implications”. So, the reader knows what’s coming next.

For Section 3, authors should introduce their proposed research framework more effective, i.e., some essential brief explanation vis-à-vis the text with a total research flowchart or framework diagram for each proposed algorithm to indicate how these employed models are working to receive the experimental results. It is difficult to understand how the proposed approaches are working.

For Section 4, authors should use more alternative models as the benchmarking models, authors should also ensure the superiority of the proposed approach, i.e., how could authors ensure that their results are superior to others? Meanwhile, authors also have to provide some insight discussion of the results. Authors can refer the following references for conducting statistical test.

Forecasting short-term electricity load using hybrid support vector regression with grey catastrophe and random forest modeling. Utilities Policy, 2021, 73, 101294.

Author Response

pls find responses to reiviewers report. Thank you for giving valuable inputs for improving our paper..

Reviewer 2 Report

Dear Authors, thank you for submitting your work and for the opportunity to review it.

I have a few sparse but detailed comments.

First, what is the difference for you between deep learning and convolutional networks.  I believe that in your work this distinction has been blurred and it is worth describing the differences.

Second, what was the exact number of teaching cases? There is information about 40000 examples and 1000 scenes. Please tell what the collection contained, in what proportion ect.

You wrote that you use LIDAR technology. I know that these are complicated and very taxing devices. In relation to the above, how many images have you collected? What is the optimal resolution? Did you also conduct research in real time and not only in virtual reality?

I also suggest rebuilding your paper to a typical layout: introduction, purpose and scope of the paper, material and methods, results, discussion, and conclusion. The one you presented is correct but unusual. I also urge you to add discussion.

The overall content grade of the paper is good, but please restructure it, clarify it, and add missing information.

Author Response

Pls find responses to reviewers suggestions. Thank you for giving oppotunity to improve our manuscript.

Round 2

Reviewer 1 Report

Authors have completely addressed all my concerns.

Reviewer 2 Report

article suitable for publication